# HEJ-Robust: A Robustness Benchmark for LLM-based Automated Program Repair

## Abstract

Recent Large Language Models (LLMs) have shown strong performance on automated program repair across standard benchmarks. However, these benchmarks evaluate models on a single canonical form of buggy code and do not reflect the syntactic variations commonly observed in real-world software, leaving robustness largely unexamined. In this work, we construct HEJ-Robust, a robustness benchmark built from HumanEval-Java-Bug using eight semantic-preserving code transformations, resulting in 1,350 transformed instances. We evaluate five fine-tuned LLMs on this benchmark and show that model performance drops by over 50% under several transformations, indicating that current LLM-based repair models lack robustness to minor syntactic variations.

## Keywords

Large Language Models, Automated Program Repair, Benchmark, Robustness Testing

### ACM Reference Format:
Anonymous Author(s). 2026. HEJ-Robust: A Robustness Benchmark for LLM-based Automated Program Repair. In . ACM, New York, NY, USA, 4 pages. https://doi.org/10.1145/nnnnnnn.nnnnnnn

## 1 Introduction

Automated program repair (APR) aims to automatically generate patches that fix buggy programs. Early APR approaches primarily followed the generate-and-validate paradigm, where candidate patches are synthesized using predefined or learned repair operators and validated against test suites. Representative systems include GenProg [21], PAR [19], and systematic mutation-based repair techniques [20, 29]. While these approaches have demonstrated effectiveness on specific bug classes, they often suffer from scalability limitations and test-suite overfitting.

Recent advances in deep learning have significantly reshaped APR research by formulating program repair as a code translation problem, where buggy code is translated into its fixed version [17, 49]. Pre-trained LLMs, such as PLBART [1], and CodeT5 [37], have shown strong repair capability when fine-tuned on bug-fix data [6, 16, 46]. More recent studies further explore instruction-tuned and agent-based LLMs for automated program repair [4, 11, 47]. To evaluate these approaches, existing benchmarks commonly rely on

defect4j [18] or HumanEval-Java-Bug [16], which assume a fixed syntactic representation of buggy programs.

Existing APR benchmarks evaluate repair accuracy on a single canonical buggy program, ignoring syntactic diversity among semantically equivalent code. Prior studies show neural code models are sensitive to semantics-preserving transformations [30, 36]. While robustness testing via transformations, fuzzing, and adversarial examples has been studied in other SE tasks [28, 43], robustness evaluation for LLM-based APR remains largely unexplored, with no standardized benchmark available.

We address this gap by introducing a transformation-based robustness benchmark for automated program repair. Constructed by applying eight semantic-preserving transformations to HumanEval-Bug [16], our benchmark enables controlled evaluation of repair consistency. We use it to assess the robustness of five fine-tuned LLM repair models against code perturbations.

The contributions of this paper are as follows:

(1) We introduce the first transformation-based robustness benchmark tailored for automated program repair.
(2) We provide a systematic evaluation of LLM-based repair models under semantics-preserving transformations.
(3) We release the benchmark to facilitate future research on robust and reliable automated program repair.

Our Code, dataset, and Artifacts are publicly available [1]

## 2 Related Work

Automated program repair (APR) has been extensively studied over the past two decades. Early work primarily follows the generate-and-validate paradigm, where candidate patches are generated and validated against test suites [15, 21, 29, 39, 42, 44]. While effective on curated benchmarks such as Defects4J [18] and QuixBugs [23], these approaches suffer from overfitting and scalability issues [12, 22, 40, 41].

More recently, deep learning-based APR approaches reformulate bug fixing as a neural machine translation problem, translating buggy code into fixed code [2, 7, 10, 14, 17, 25, 35, 49]. Pre-trained LLMs further improve repair performance by leveraging large-scale code corpora before fine-tuning on repair data [1, 6, 8, 13, 24, 26, 37, 46]. Most of these approaches evaluate on bug-fix pairs (BFPs) [8, 35], which largely consist of abstract or canonicalized code. More recent benchmarks derived from HumanEval [9] enable functional validation using test cases [16]. Complementary studies explore LLM-based repair in competitive programming and agent-based settings [4, 11, 47].

Parallel to APR research, robustness testing of neural models for code has gained attention. Prior work demonstrates that neural code models are vulnerable to small, semantics-preserving transformations [3, 33, 48]. Transformation-based testing, fuzzing, and

---

[1]https://github.com/anonprox/hej-robust

**Table 1: Fine-tuned models against different semantic-preserving code transformations on HumanEval-Java.**

**(a) Local Variable Renaming (100 bugs)**

| | Pass@10 | | | CodeBLEU | |
| --- | --- | --- | --- | --- | --- |
| | orig. | trans. | change | orig. | trans. |
| plbart_base | 14.53 | 6.54 | 54.99↓ | 82.11 | 81.91 |
| plbart_large | 21.88 | 9.91 | 54.71↓ | 82.75 | 82.15 |
| codet5_small | 19.35 | 8.26 | 57.31↓ | 82.17 | 81.43 |
| codet5_base | 24.81 | 12.28 | 50.5↓ | 82.02 | 81.58 |
| codet5_large | 23.66 | 11.5 | 51.39↓ | 80.74 | 80.92 |

**(b) Method Renaming (149 bugs)**

| | Pass@10 | | | CodeBLEU | |
| --- | --- | --- | --- | --- | --- |
| | orig. | trans. | change | orig. | trans. |
| plbart_base | 19.46 | 18.13 | 6.83↓ | 82.96 | 82.78 |
| plbart_large | 23.98 | 22.8 | 4.92↓ | 83.34 | 83.04 |
| codet5_small | 21.16 | 19.46 | 8.03↓ | 83.1 | 82.96 |
| codet5_base | 25.87 | 24.37 | 5.8↓ | 83.06 | 82.71 |
| codet5_large | 24.75 | 23.98 | 3.11↓ | 81.73 | 81.69 |

**(c) Parameter Renaming (162 bugs)**

| | Pass@10 | | | CodeBLEU | |
| --- | --- | --- | --- | --- | --- |
| | orig. | trans. | change | orig. | trans. |
| plbart_base | 17.86 | 18.69 | 4.65↑ | 83.63 | 83.84 |
| plbart_large | 22.6 | 23.33 | 3.23↑ | 84.11 | 83.86 |
| codet5_small | 20.3 | 19.1 | 5.91↓ | 83.92 | 84.02 |
| codet5_base | 24.77 | 24.41 | 1.45↓ | 83.86 | 83.94 |
| codet5_large | 24.41 | 23.7 | 2.91↓ | 82.67 | 82.55 |

**(d) Boolean Exchange (7 bugs)**

| | Pass@10 | | | CodeBLEU | |
| --- | --- | --- | --- | --- | --- |
| | orig. | trans. | change | orig. | trans. |
| plbart_base | 12.5 | 22.22 | 77.76↑ | 86.03 | 85.8 |
| plbart_large | 22.22 | 30.0 | 35.01↑ | 86.14 | 85.69 |
| codet5_small | 22.22 | 22.22 | 0% | 83.68 | 83.3 |
| codet5_base | 22.22 | 12.5 | 43.74↓ | 86.02 | 85.47 |
| codet5_large | 12.5 | 12.5 | 0% | 85.26 | 84.67 |

**(e) Loop Exchange (142 bugs)**

| | Pass@10 | | | CodeBLEU | |
| --- | --- | --- | --- | --- | --- |
| | orig. | trans. | change | orig. | trans. |
| plbart_base | 19.32 | 18.39 | 4.81↓ | 84.66 | 84.76 |
| plbart_large | 25.26 | 23.66 | 6.33↓ | 84.77 | 85.5 |
| codet5_small | 21.55 | 17.92 | 16.84↓ | 84.08 | 84.61 |
| codet5_base | 26.04 | 23.66 | 9.14↓ | 84.58 | 85.12 |
| codet5_large | 28.28 | 26.8 | 5.23↓ | 83.29 | 84.36 |

**(f) Reorder Condition (603 bugs)**

| | Pass@10 | | | CodeBLEU | |
| --- | --- | --- | --- | --- | --- |
| | orig. | trans. | change | orig. | trans. |
| plbart_base | 16.88 | 15.69 | 7.05↓ | 83.83 | 85.64 |
| plbart_large | 21.41 | 18.48 | 13.69↓ | 84.17 | 86.1 |
| codet5_small | 19.7 | 17.92 | 9.04↓ | 84.01 | 85.87 |
| codet5_base | 23.25 | 20.99 | 9.72↓ | 83.94 | 85.75 |
| codet5_large | 23.45 | 21.62 | 7.8↓ | 82.56 | 85.04 |

**(g) Insert Log Statement (173 bugs)**

| | Pass@10 | | | CodeBLEU | |
| --- | --- | --- | --- | --- | --- |
| | orig. | trans. | change | orig. | trans. |
| plbart_base | 17.22 | 16.43 | 4.59↓ | 83.64 | 83.6 |
| plbart_large | 22.07 | 22.42 | 1.59↑ | 84.1 | 83.85 |
| codet5_small | 19.53 | 18.4 | 5.79↓ | 83.86 | 83.69 |
| codet5_base | 24.45 | 22.07 | 9.73↓ | 83.8 | 83.57 |
| codet5_large | 24.78 | 24.45 | 1.33↓ | 82.61 | 82.91 |

**(h) Insert Try catch (114 bugs)**

| | Pass@10 | | | CodeBLEU | |
| --- | --- | --- | --- | --- | --- |
| | orig. | trans. | change | orig. | trans. |
| plbart_base | 16.91 | 13.74 | 18.75↓ | 83.86 | 83.65 |
| plbart_large | 21.53 | 19.29 | 10.4↓ | 84.17 | 83.94 |
| codet5_small | 19.29 | 11.02 | 42.87↓ | 84.52 | 84.49 |
| codet5_base | 25.17 | 18.12 | 28.01↓ | 84.29 | 84.3 |
| codet5_large | 26.14 | 18.12 | 30.68↓ | 83.16 | 83.52 |

adversarial example generation have been applied to code models [28, 31, 38, 45], with later work emphasizing natural and context-aware transformations [43]. While robustness has been studied for tasks such as code summarization and code representation learning, it remains largely unexplored for automated program repair. In particular, existing APR benchmarks do not systematically evaluate the robustness of repair models under semantics-preserving code transformations.

In this work, we bridge this gap by focusing on robustness benchmarking for LLM-based program repair rather than proposing a new repair technique.

## 3 Robustness Benchmark Design

### 3.1 Base Dataset

We adopt the HumanEval-Java-Bug dataset introduced by Jiang et al. [16], which is derived from HumanEval [9]. The dataset contains 164 Java programs with manually injected bugs and annotated buggy-line locations. Each instance is accompanied by executable

test cases and human-written patches. We select this dataset because it is manually curated, recent, and less likely to suffer from data leakage issues common in earlier APR benchmarks.

## 3.2 Semantic-Preserving Code Transformations

We apply eight semantic-preserving code transformations that reflect common syntactic variations observed in real-world software. Prior work has shown that such transformations are effective for evaluating the robustness of neural code models. Our goal is not to improve code quality, but to assess robustness under benign syntactic changes.

The eight transformations are: (1) local variable renaming, (2) method renaming, (3) parameter renaming, (4) log statement insertion, (5) try–catch insertion, (6) boolean exchange, (7) loop exchange, and (8) condition reordering. All transformations preserve program semantics and do not change test outcomes.

**Renaming transformations (1–3).** For identifier renaming, we adopt the naturalness-aware substitution strategy proposed by Yang et al. [43]. Unlike prior approaches that use random strings or fixed patterns [28, 31, 38, 45], this method generates context-aware and developer-natural identifiers, ensuring that performance degradation reflects robustness issues rather than unnatural code artifacts.

We use masked language prediction with CodeBERT and Graph-CodeBERT to generate candidate identifiers and select substitutions based on cosine similarity in embedding space. Java code is parsed using tree-sitter [5] to ensure consistent replacement across all occurrences. To control transformation strength, only one identifier is renamed per program.

**Structural and syntactic transformations (4–8).** The remaining transformations are implemented using JavaTransformer [32], which applies AST-based modifications via JavaParser. Logging statements are inserted at method entry points, and try–catch blocks are added at syntactically valid locations. Boolean exchange inverts boolean initializations while preserving semantics. Loop exchange converts `for` loops to equivalent `while` loops and vice versa. Condition reordering swaps operands in equality and inequality expressions. Transformations are applied only when syntactically valid.

All transformations preserve semantics and syntax, yielding 1,350 instances.

## 3.3 Benchmark Construction and Task Formulation

After applying transformations, the locations of buggy lines may change. We manually re-annotate the buggy-line locations for all transformed programs. Combined with the original human-written patches and test cases, this yields a fully executable benchmark suitable for robustness evaluation. Model outputs are evaluated using both code-similarity metrics, such as CodeBLEU [27, 34], and functional correctness via test-based metrics (e.g., pass@10) provided by HumanEval-Java-Bug [16].

## 4 Experimental Setup

To evaluate the proposed benchmark, we consider five LLMs: two PLBART variants and three CodeT5 variants, all fine-tuned and

released by Jiang et al. [16]. We directly evaluate these models without further modification. We use CodeBLEU as a similarity-based metric, while Pass@10 serves as the primary indicator of functional correctness. Results are compared to quantify robustness degradation under semantic-preserving transformations, as reported in Table 1. More results with pre-trained models are available in our shared repository.

## 5 Results

The evaluation results are summarized in Table 1, which reports the performance of five fine-tuned models across eight transformed datasets. Each transformation is presented in a separate subtable, showing Pass@10 and CodeBLEU scores for both the original and transformed datasets. We also report the relative change from the original to the transformed dataset, indicated by ↑ for improvements and ↓ for degradations.

Across all eight transformations, we observe consistent drops in Pass@10 for all models, with the largest degradation occurring under the Local Variable Renaming transformation, where performance decreases by 50.5% to 57.31%. Notably, robustness does not correlate with model size: larger models often degrade more than their smaller counterparts (e.g., CodeT5_large vs. CodeT5_base, and PLBART_large vs. PLBART_base) across multiple transformations. Results for Boolean Exchange are unstable due to the small number of affected samples (7 instances).

In contrast, CodeBLEU scores remain largely stable across transformations, with only minor increases or decreases. This discrepancy suggests that CodeBLEU may not fully capture functional robustness, which we leave for future investigation.

## 6 Conclusion

We present **HEJ-Robust**, a robustness benchmark of 1,350 bug instances constructed from HumanEval-Java-Bug using eight semantic-preserving transformations. Evaluating five fine-tuned LLMs, we show that even minor syntactic variations cause consistent drops in Pass@10, revealing substantial robustness gaps in current repair models. Future work will study richer transformations, larger contexts, and improved robustness-aware training and evaluation metrics.

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
