# OpenReview forum: "HEJ-Robust: A Robustness Benchmark for LLM-based Automated Program Repair"
_ACM.org/AIWare/2026/Conference — Submitted to AIware 2026_

### Official Review · Reviewer_gmrG · 2026-02-21

**Rating:** 2
**Confidence:** 4

**Review:**

### Strengths

* This paper targets a concrete and under-evaluated aspect of LLM-based APR: robustness to harmless syntactic variation in buggy codes, using a clear transformation-based setup.

### Weaknesses

1. Limited novelty and incomplete related work. Closely related transformed APR datasets and studies (e.g., DEFECTS4J-TRANS) are missing, which makes the “first” claim difficult to defend and leaves the contribution unclear.
2. Benchmark construction and validation are under-specified. It is not clear what guarantees are provided that transformations preserve semantics beyond passing the given tests, and how manual bug-line relabeling was validated.
3. The experimental scope is narrow for the claimed conclusions. Evaluation is limited to 5 older fine-tuned models and one base dataset, with small subsets for some transformations, and analysis is limited to pass@10 and CodeBLEU.

### Detailed comments

The core idea of evaluating APR under semantics-preserving transformations is not new. Defects4J-TRANS (Li et al., ICSE 2025 NIER, arXiv:2503.09217) is directly relevant and should be discussed. Without this, the paper reads as an incremental dataset variant rather than a new benchmark direction. The authors should clearly articulate what is novel here relative to prior transformed datasets, including the choice of base dataset, the specific transformation suite, the renaming strategy, the annotation protocol, and any new insights enabled by the benchmark. If the intent is to claim “first” for HumanEval-Java-Bug, that should be stated precisely and scoped accordingly.

A robustness benchmark is only as strong as its guarantees. The paper states that transformations are semantics-preserving, but the validation procedure is not described at the level needed for trust. Passing the existing tests is a weak proxy because the tests may be underspecified. If additional checks were used (compilation checks, differential execution, static constraints, or transformation-specific invariants), they should be explicitly listed. Likewise, manual relabeling of buggy-line locations is a potential source of noise and bias, and the paper should describe how correctness was ensured (independent verification, spot checks, or consistency rules).

The reported performance drops are plausible, but the experimental evidence is limited. Some transformation categories have too few instances to support strong claims, and the models are too old to be investigated effectively. The paper would be stronger with at least one modern model or sota APR pipeline.

### Questions

1. How do you verify that each transformation is semantics-preserving beyond compilation and the provided tests?

2. The abstract reports 1,350 transformed instances, but Table 1 sums to 1,450. Which number is correct, and why?

3. How does HEJ-Robust differ from Defects4J-TRANS and other transformation-based APR generalizability datasets, and how should the novelty claim be revised accordingly?

**Summary:**

This paper presents HEJ-Robust, a robustness benchmark derived from HumanEval-Java-Bug by applying 8 semantics-preserving source code transformations (renaming, statement insertion, and equivalent rewrites). The goal is to test whether LLM-based automated program repair models remain effective under benign syntactic variation. The authors evaluate 5 fine-tuned models (PLBART and CodeT5 variants), and report drops in pass@10 under several transformations, while CodeBLEU remains relatively stable.

Overall, the topic is relevant, but the submission does not meet the bar for novelty and rigor expected at a software engineering venue. Similar transformation-based datasets for APR generalizability already exist (e.g., Defects4J-TRANS from Li et al., ICSE 2025 NIER), yet they are not cited, and the paper’s “first” claim becomes hard to justify. The dataset construction and reporting also have internal inconsistencies that weaken credibility.

---

> ### Author Response · Authors · 2026-03-16
>
> We thank the reviewer for the helpful feedback. Below, we clarify the novelty, transformation validity, annotation process, and experimental scope of HEJ-Robust.
>
> **Novelty compared with Defects4J-TRANS
>
> We agree that Defects4J-TRANS is closely related and should be discussed. We will include this work in the related work section and clarify the differences. HEJ-Robust introduces five additional transformations beyond Defects4J-TRANS, including method renaming, parameter renaming, condition reordering, log statement insertion, and try–catch insertion, creating a broader set of syntactic variations. In addition, our benchmark is built on HumanEval-Java-Bug, a human-crafted dataset with function-level tasks, whereas Defects4J uses project-level real bugs. This provides a complementary evaluation setting for studying the robustness of LLM-based program repair models. We will also revise the novelty claim to clearly scope it to HumanEval-Java-Bug–based robustness evaluation.
>
> **Transformation validity
>
> We directly use two semantic-preserved transformation tools from established prior works: renaming transformations from [1] and structural/syntactic transformations from [2], both designed to preserve program semantics. These transformation tools have been widely validated in prior research, and we follow the same transformation procedures in our implementation. Therefore, we rely on their semantics-preserving guarantees in constructing the benchmark.
>
> **Bug-line annotation after transformation
>
> After transformations are applied, code line positions may change. To maintain correct repair localization, the authors manually re-annotated the buggy-line locations by inspecting each transformed program and identifying the new location corresponding to the original buggy statement. Instances where transformations modified the buggy line itself were removed from the dataset.
>
> **Instance count inconsistency
>
> The correct number is 1,450 transformed instances, as reflected in Table 1. The value 1,350 in the text is a typographical error and will be corrected in the revision.
>
> **Experimental scope
> The PLBART and CodeT5 models were fine-tuned by Jiang et al. [3] specifically for APR tasks and evaluated on the HumanEval-Java-Bug benchmark. Using these models allows us to study the robustness of APR-specific fine-tuned models under semantic-preserving transformations. To broaden the experimental scope, we will extend the evaluation to include modern code models such as Magicoder-7B and CodeLlama-7B in the revision.
>
> 1. Li, Fengjie, et al. "Evaluating the generalizability of llms in automated program repair." 2025 IEEE/ACM 47th International Conference on Software Engineering: New Ideas and Emerging Results (ICSE-NIER). IEEE, 2025.
> 2. Rabin, Md Rafiqul Islam, et al. "On the generalizability of neural program models with respect to semantic-preserving program transformations." Information and Software Technology 135 (2021): 106552.
> 3. Jiang, Nan, et al. "Impact of code language models on automated program repair." 2023 IEEE/ACM 45th International Conference on Software Engineering (ICSE). IEEE, 2023.
> 4. Wang, Shiqi, et al. "ReCode: Robustness evaluation of code generation models." Proceedings of the 61st Annual Meeting of the Association for Computational Linguistics (Volume 1: Long Papers). 2023.

---

> > ### Comment · Reviewer_gmrG · 2026-03-19
> >
> > Thank you for the detailed response. I appreciate the clarifications regarding the relation to Defects4J-TRANS, the corrected instance count, and the planned revision of the novelty claim. The problem itself is meaningful, and I agree that HumanEval-Java-Bug provides a complementary setting for studying robustness in APR. However, my main concerns about novelty relative to prior work, the strength of semantics preservation and annotation validation, and the limited experimental scope are only partially addressed in the current response. Thus, I will keep my current assessment unchanged, and I appreciate the effort behind the response.

---

> > > ### Author Response · Authors · 2026-03-20
> > >
> > > We thank the reviewer for the detailed follow-up and for the constructive feedback. We understand the reviewer’s concerns regarding novelty, transformation validity, annotation, and experimental scope, and we clarify them below.
> > >
> > > Novelty and positioning. We plan to revise the paper to clearly scope our contribution as a robustness benchmark on HumanEval-Java-Bug, focusing on function-level, human-crafted bugs, and clarify its distinction from Defects4J-TRANS in terms of both dataset setting and transformation coverage. In particular, our benchmark includes five additional transformations beyond Defects4J-TRANS: method renaming, parameter renaming, condition reordering, log statement insertion, and try–catch insertion, enabling a broader range of syntactic variations.
> > >
> > > Transformation validity. The transformations are implemented using established tools from prior work [1,2] that are designed to preserve program semantics. We follow these standard procedures and plan to clarify this more explicitly, along with related limitations.
> > >
> > > Bug-line annotation. After applying transformations, buggy-line locations are manually re-annotated by inspecting the transformed code and mapping the original buggy statement to its updated position. Instances where transformations directly affect the buggy line are removed. We plan to clarify this annotation process more explicitly in the revision.
> > >
> > > Experimental scope. We use APR-specific fine-tuned models (PLBART and CodeT5) from prior work [3] to evaluate robustness under transformations. To improve coverage, we plan to extend the evaluation with additional 2 new models (e.g., Magicoder-7B and CodeLlama-7B) which are currently running.

---

> > > > ### Comment · Reviewer_gmrG · 2026-03-21
> > > >
> > > > Thank you for the follow-up clarification. I appreciate the effort to better position the paper and to more precisely scope the contribution to HumanEval-Java-Bug. However, this response still mainly points to planned revisions and additional experiments rather than evidence in the current submission, so it does not materially change my assessment. I will keep my current assessment unchanged.

---

### Official Review · Reviewer_qqtv · 2026-03-08

**Rating:** 2
**Confidence:** 4

**Review:**

# Strength
+ The problem is important, targeting the robustness of the AVR system
+ Proposed approach is promising

# Weakness
+ Some technical design requires more justification or details
+ Some important related works are missing
+ Selection of baselines is limited

# Detailed Comments
- Not all listed transformations are obviously semantics-preserving across contexts.
- Limited model set (older fine-tuned PLBART/CodeT5 variants) and a single dataset constrain generality; evaluation on stronger modern LLMs (e.g., CodeLlama, StarCoder2, Qwen2.5-Coder) in an APR setting would increase relevance.
- Missing details about decoding (beam vs. sampling), seeds, number of generations for Pass@10, and whether multiple runs are averaged; no statistical significance tests are reported.
- While several robustness papers are cited, many recent relevant papers are missing, e.g., Recode (ReCode: Robustness Evaluation of Code Generation Models).

**Summary:**

The paper introduces HEJ-Robust, a robustness benchmark for LLM-based automated program repair constructed by applying eight semantics-preserving code transformations to HumanEval-Java-Bug.
This paper re-annotates buggy-line locations for transformed instances and evaluates five fine-tuned LLMs using Pass@10 and CodeBLEU, observing substantial drops in functional correctness under several transformations.

---

> ### Author Response · Authors · 2026-03-16
>
> We thank the reviewer for the helpful feedback. Below, we address the concerns regarding transformations, models, experimental setup, and related work.
>
> **Semantics-preserving transformations
> The transformations are implemented using two established prior works: renaming transformations from [1] and structural/syntactic transformations from [2], both designed to preserve program semantics. We directly adopt these transformation techniques in our implementation. Since these methods have been widely validated in prior research, we rely on their semantics-preserving properties in our study.
>
> **Baseline model selection
> The PLBART and CodeT5 models were fine-tuned by Jiang et al. [3] specifically for APR tasks and evaluated on the HumanEval-Java-Bug benchmark [3]. Using these models allows us to evaluate robustness for APR-specific fine-tuned models under semantic-preserving transformations. We plan to extend the evaluation with additional models such as Magicoder-7B and CodeLlama-7B, which are currently running.
>
> **Experimental setup details
> We directly used the fine-tuned models released by prior work [3] and applied them to our transformed dataset without modifying the original training or decoding configurations. Following the evaluation protocol in [3], we generate 10 candidate patches per buggy program and evaluate them using the pass@10 metric, where a bug is considered fixed if at least one generated patch passes all developer-written test cases.
> Regarding randomness, we did not modify the seed settings of the released models and followed the original generation procedure provided in [3]. We will clarify the generation configuration, seed handling, and evaluation procedure in the revision.
>
> **Related work
> We agree that additional related work should be discussed. We will expand the related work section to include recent robustness studies such as ReCode [4].
>
> 1. Li, Fengjie, et al. "Evaluating the generalizability of llms in automated program repair." 2025 IEEE/ACM 47th International Conference on Software Engineering: New Ideas and Emerging Results (ICSE-NIER). IEEE, 2025.
> 2. Rabin, Md Rafiqul Islam, et al. "On the generalizability of neural program models with respect to semantic-preserving program transformations." Information and Software Technology 135 (2021): 106552.
> 3. Jiang, Nan, et al. "Impact of code language models on automated program repair." 2023 IEEE/ACM 45th International Conference on Software Engineering (ICSE). IEEE, 2023.
> 4. Wang, Shiqi, et al. "ReCode: Robustness evaluation of code generation models." Proceedings of the 61st Annual Meeting of the Association for Computational Linguistics (Volume 1: Long Papers). 2023.

---

> > ### Comment · Reviewer_qqtv · 2026-03-21
> >
> > Thank the authors for their response. However, the core concern remains unresolved: the paper's main contribution is a robustness benchmark, yet the evaluation is limited to older, smaller fine-tuned models and lacks sufficient experimental rigor, and the promised extensions to stronger models are not yet part of the submission. I'd like to maintain my original score.

---

### Official Review · Reviewer_9UoC · 2026-03-11

**Rating:** 3
**Confidence:** 4

**Review:**

Thank you for submitting to AIWare 2026. The idea of studying robustness under syntactic variations is interesting, as real-world software often exhibits multiple syntactic forms for semantically equivalent programs. The paper is generally easy to follow, but several aspects would benefit from further clarification.

First, the choice of semantic-preserving transformations requires stronger justification. While the paper introduces eight transformations (e.g., variable renaming, loop exchange, logging insertion), it is unclear to what extent these reflect common variations in real-world software development. Providing empirical evidence or references on their representativeness would strengthen the benchmark's external validity. At minimum it could be explicitly discussed in the threat-to-validity section.

Second, the evaluation considers a relatively limited set of models, focusing mainly on several fine-tuned PLBART and CodeT5 variants. It would be helpful to justify this selection more clearly. Why were these particular models chosen? Would the observed robustness issues persist for other model families or more recent APR approaches, such as prompt-based or agent-based repair systems? Without considering a broader range of models, the generality and significance of the findings may be somewhat limited.

Third, the robustness analysis could be expanded. The paper mainly reports Pass@10 and CodeBLEU, showing that functional correctness degrades under certain transformations. However, the discussion of these results remains relatively brief. Why do certain transformations lead to significantly larger degradation than others? What types of repair failures occur under these transformations? A deeper analysis of these behaviors would provide more insight into the robustness limitations of current APR models and strengthen the overall contribution of the benchmark.

**Summary:**

The paper introduces a benchmark named HEJ-Robust to evaluate the robustness of LLM-based automated program repair models under semantic-preserving code transformations. The benchmark is constructed from the HumanEval-Java-Bug dataset by applying eight transformations that alter the syntax of buggy programs while preserving their semantics. The authors evaluate several fine-tuned APR models on this benchmark and find that repair performance drops significantly under certain transformations, revealing that current LLM-based repair systems are sensitive to minor syntactic variations.

---

> ### Author Response · Authors · 2026-03-16
>
> Thank you for the constructive feedback and positive comments about the importance of studying robustness under syntactic variations.
>
>
> **Representativeness of transformations
>
> The transformations used in HEJ-Robust are derived from two established prior works: renaming transformations from [1] and structural/syntactic transformations from [2]. These transformations correspond to common code variations seen in practice, such as identifier renaming, condition reordering, and insertion of logging or try–catch blocks, which are typical refactoring or maintenance operations in software development. Since these transformation tools have been widely used and validated in prior research, we adopt the same procedures in our benchmark construction. We will further clarify their representativeness and discuss potential limitations in the Threats to Validity section in the revision.
>
>
> **Model selection
>
> The PLBART and CodeT5 models were fine-tuned by Jiang et al. [3] specifically for APR tasks and evaluated on the HumanEval-Java-Bug benchmark [3]. Using these models allows us to study the robustness of APR-specific fine-tuned models on our transformed dataset. We plan to extend the evaluation by including additional models such as Magicoder-7B and CodeLlama-7B, which are currently running. We will add the justification of selecting the target models in the revision.
>
>
> **Analysis of robustness degradation
>
> Transformations such as identifier renaming introduce distribution shifts in variable names and code structure, which can disrupt token patterns learned during model training. As a result, repair performance (Pass@10) decreases even though program semantics remain unchanged. A likely reason is that models rely heavily on identifier patterns learned during training, making them sensitive to variable-name changes. Similar behavior is also reported in ReCode [4], where variable renaming causes larger degradation than other transformations. We will include more in-depth analysis of these effects in the revision.
>
> 1. Li, Fengjie, et al. "Evaluating the generalizability of llms in automated program repair." 2025 IEEE/ACM 47th International Conference on Software Engineering: New Ideas and Emerging Results (ICSE-NIER). IEEE, 2025.
> 2. Rabin, Md Rafiqul Islam, et al. "On the generalizability of neural program models with respect to semantic-preserving program transformations." Information and Software Technology 135 (2021): 106552.
> 3. Jiang, Nan, et al. "Impact of code language models on automated program repair." 2023 IEEE/ACM 45th International Conference on Software Engineering (ICSE). IEEE, 2023.
> 4. Wang, Shiqi, et al. "ReCode: Robustness evaluation of code generation models." Proceedings of the 61st Annual Meeting of the Association for Computational Linguistics (Volume 1: Long Papers). 2023.

---

### Author Response · Authors · 2026-03-18

Dear Reviewers, we have submitted our responses to all three reviews. We kindly request that you take a moment to review our responses and update your assessments if appropriate. We are happy to provide any further clarification. Thank you for your time and consideration.